# Heating and Compression at Elevated Temperature of Thin-Walled Titanium Channel Section Columns

**DOI:** 10.3390/ma14112928

**Published:** 2021-05-29

**Authors:** Adrian Gliszczyński, Leszek Czechowski, Nina Wiącek

**Affiliations:** Department of Strength of Materials, Faculty of Mechanical Engineering, Lodz University of Technology, Stefanowskiego 1/15, 90-924 Lodz, Poland; leszek.czechowski@p.lodz.pl (L.C.); nina.wiacek@dokt.p.lodz.pl (N.W.)

**Keywords:** stability, thermal buckling, titanium, isotropic hardening, thin-walled structures

## Abstract

The paper deals with numerical and experimental investigations of the channel section column subjected to heating and compression at elevated temperature. The analyzed columns were made of titanium alloy (Grade 2) and simply supported on both ends. The research procedure involved initial compression of the column (i), heating the preloaded column (ii) and compression of the column at elevated temperature to failure (iii). The tests were performed at temperatures from 23 °C to 300 °C. Numerical calculations were carried out in the Ansys^®^ software and involved the application of bilinear and multilinear isotropic hardening. It has been revealed that the temperature increase in a statically indeterminate system causes a decrease in the load-carrying capacity of the profile. An increase in temperature by 27 °C causes a reduction of the load-carrying capacity by 10%, while compression at temperature 300 °C reduces the nominal load-carrying capacity of the profile by half. Most of the proposed numerical procedures allowed for accurate estimation of reaction forces during heating and maximum compressive forces recorded during compression at elevated temperatures. The correctness of the determined material characteristics and the suitability of shell models for estimation of the response of a thin-walled structure subjected to thermomechanical loading was confirmed.

## 1. Introduction

Titanium structures and members regarding their moderately high-temperature resistance, high strength and good corrosion resistance are designed to work in untypical conditions [1,2]. Titanium alloys Ti-6Al-4V are distinguished by a super plasticity occurring in temperature of 650–750 °C [3]. The essential feature of titanium is its biocompatibility because this material and its alloys are often used in medicine branches as biomaterials to bear heavy loads [4]. As far as thermal analysis is concerned, there are papers devoted to determination of mechanical materials properties of samples at elevated temperature due to tension [5,6]. Compression of welded S460 steel columns in temperature field was examined in [7]. The research on metal beams strength at higher temperature was performed in references [8,9]. Jiun et al. [10,11] investigated experimentally thermal buckling of circular laminated composite and aluminum plates by means of Digital Image Correlation (DIC) technique. Nguyena et al. [12] examined the behavior of the carbon fiber reinforced polymer (CFRP) structures under both temperature environment and mechanical loads. Zhou et al. [13] investigated mechanical properties of CFRP composites in a temperature field. Glassman et al. [14] investigated the mechanical properties of steel A588 at temperature coming up to 815 °C by taking into consideration Young’s modulus, yield stress, fracture toughness, ultimate stress and surface hardness. Khaneghahi et al. [15] investigated the behavior of glass fiber reinforced polymer (GFRP) profiles compressed at higher temperature. Jo et al. [16] examined the buckling of compressed plate structure at high temperature. Zhang et al. [17] demonstrated the mechanical properties of IN718 alloy manufactured by laser metal deposition at elevated temperature. Another studies of material properties of high strength aluminum alloys, cold-rolled steel, high-chromium austenitic stainless steel and high strength steel at elevated temperature are included in [18,19,20,21], respectively. Wang et al. in [22] analyzed the buckling of restrained steel Q460 columns subjected to high temperature according to fire resistant standard ISO-834 and investigated the creep buckling of columns at elevated temperature in [23]. Theoretical approaches involving thermal buckling of functionally graded structures can be found in [24,25,26,27]. Among others, the failure of thin-walled plate structures relied upon theoretical and experimental approach was also analyzed in papers [28,29,30,31,32,33]. These works are strictly associated with phenomenon of stability occurring in thin-walled structures.

Although in the literature there are papers devoted to the behavior of thin-walled structures at elevated temperatures, the vast majority of these publications refer to the influence of open fire on the structure. However, there are no studies devoted to the response of thin-walled structures subjected to elevated temperatures in the range of 200–450 °C. This is not surprising, since in formal terms in the range of these temperatures isotropic materials do not show significant differences from the lower temperatures. However, this work shows that in statically indeterminate systems, a column made of isotropic titanium alloy at a temperature of 300 °C reduces its load capacity by half in relation to the response recorded at ambient temperature. It should be highlighted that the undertaken research was devoted to simply supported titanium (Grade 2) columns tested at temperatures ranging from 23 °C to 300 °C (every 25 °C). In order to achieve a statically indeterminate state the experimental research was divided into three stages: initial compression of the column (i), heating the preloaded column with blocked supports (ii) and compression of the column at elevated temperature up to failure (iii). Due to the surprising effect of the experiment, it was decided to systematically study the above phenomenon using the finite element method. To validate numerical simulations experimental campaign was performed. Numerical calculations were based on implementation of bilinear or multilinear characteristics of isotropic hardening. Despite the fact that the material characteristics for the analyzed temperatures were provided, the analysis of the influence of the assumed tangential modulus, the buckling mode and amplitude of initial imperfections was also conducted. The Digital Image Correlation system (Aramis^®^ [34]) was employed to register the deformations of columns at each stage of the test. The paper shows essential influence of mechanical properties of titanium on stability and load-carrying capacity of channel section profiles at elevated temperatures.

## 2. Considered Plasticity Models

In general, plasticity is used to reflect the behavior of materials overloaded beyond their elastic limit. The considered titanium as well as the other metals have an initial elastic range represented by the linear relationship between deformation and load. However, it should be kept in mind that beyond the elastic limit a nonrecoverable plastic strain develops. Thus, complete removal of the load may result in a permanent deformation due to the presence of plastic strain in the material. In general, evolution of the plastic strain is based on the load history. The main factors contributing to the evolution of the plastic zone are temperature, stress and/or strain rate. Internal variables such as yield strength, back stress and damage are no less important. The constitutive equations intended to reflect elastic-plastic behavior assume decomposition of the total strain into elastic and plastic parts [35,36]. The main elements of the plastic constitutive models are:the yield criterion that defines the material state at the transition from elastic to elastic-plastic behavior;the flow rule that determines the increment in plastic strain from the increment in load;the hardening rule that gives the evolution in the yield criterion during plastic deformation.

In the undertaken research two isotropic hardening models were used: bilinear and multilinear. Both considered hardening models assume a von Mises yield criterion. The criterion is isotropic and can be written in following form Equation (1):f(σ, σ_0_) = σ_e_ − σ_0_ = 0(1)
where σ_e_ is the von Mises effective stress, also known as the von Mises equivalent stress Equation (2): (2)$$\sigma_{e}=\sqrt{\frac{3}{2}\left(\sigma :\sigma -\frac{1}{3}\mathrm{tr}{\left(\sigma \right)}^{2}\right)}$$
and σ_0_ is the yield strength traditionally determined from uniaxial tensile tests. In principal stress space, the yield surface is represented by a cylinder [37]. For an associated flow rule, the plastic potential is the yield criterion in Equation (2) and, according to Equation (3), the plastic strain increment is proportional to the deviatoric stress:(3)$${d\varepsilon }^{\mathrm{pl}}=d\lambda \left(\sigma -\frac{1}{3}\mathrm{tr}(\sigma )\right)$$

Bilinear isotropic hardening is represented by a bilinear effective stress vs. effective strain curve. The slope of the characteristic curve in the first stage is a representation of the nominal Young’s modulus of the material, while after exceeding the yield strength, plastic deformation is initiated. The second section of the curve is defined by the tangent modulus E* and thus represents the stiffness reduction and the evolution of plastic strains. It should be noted that the value of the tangent modulus E* cannot be less than zero or greater than the nominal elastic modulus E. Literature review proves that the tangent modulus should be in the range from one-tenth to one-thousandth of the nominal Young’s modulus [38,39,40,41]. As a rule, the multilinear model does not differ significantly from the bilinear model. The main difference lies in the approach to modeling the hardening, which in this case is represented by a piece-wise linear stress vs. total strain curve with a consistently decreasing tangent modulus. The tangent modulus in each successive interval must therefore be lower than the previous one, but always greater than zero. Both approaches are schematically presented in Figure 1.

## 3. Experimental Procedure

The columns taken into consideration were 250 mm long (L), with the following cross-section dimensions: width of the flange (H): 40 mm, width of the web (W): 80 mm and wall thickness (t): approx. 1.5 mm (cf. Figure 2b). The material of analyzed columns was titanium alloy (Grade 2) with content of almost pure titanium. The specification of the chemical composition of Grade 2 titanium alloy allows the following standards for the chemical elements: Ti > 98.9%, Fe < 0.30%, O < 0.25%, C < 0.08%, N < 0.03%, H < 0.015%. However, the certificate provided by the manufacturer (BIMO TECH (Poland)) indicates the following content of chemical elements in the analyzed material: Fe < 0.018%, O < 0.14%, C < 0.0067%, N < 0.0058%, H < 0.0022%, Ti > remainder). The experimental procedure involved three steps:initial compression of the column,heating the preloaded column,compression of the column at elevated temperature.

The first stage consisted in initial compression of the column to approx. 5–6% of the nominal load-carrying capacity of the profile. Then, the loading support of the testing machine was blocked and the sample was heated by ΔT. Apart from the tests at the ambient temperature (approx. 23 °C), the following temperatures inside the thermal chamber were considered: 50 °C, 75 °C, 100 °C, 125 °C, 150 °C, 175 °C, 200 °C, 225 °C, 250 °C, 275 °C and 300 °C. The column heating process for each test took about 2 h. The final step was the static compression of the columns to a global failure. The compression tests were carried out in the form of uniform shortening of the profiles. The velocity of the traverse was set to 1 mm/min.

### 3.1. Test Stand

The compression tests were conducted on a universal testing machine (UTM)—INSTRON 5982 (Instron, Norwood, MA, USA). The machine enables conducting the tests in range from 0.02 N to 100 kN. The titanium columns were compressed in thermal chamber—INSTRON 3119–605 (Instron, Norwood, MA, USA). The thermal chamber allows to reach temperatures of up to 350 °C. The test stand is shown in Figure 2a. Measurement data were collected directly from the UTM (force and displacement of the traverse approximately corresponding to the shortening of the profile during compression) and using the digital image correlation (DIC) system (Aramis^®^ [34]). The application of the Aramis system allowed to determine the displacement maps of the tested profiles during heating and during compression until the profile failure. Moreover, the coupling of the Aramis system with the force sensor of the UTM and the chamber temperature sensor allowed for the collection of data with a given measurement frequency. During heating, data were collected every 5 min (while the average time of heating was about 2 h) and in the case of compression after heating with a frequency of 1 Hz. It should be mentioned that the measured temperature concerned the measurement of the air inside the thermal chamber and not the actual temperature of the column. Nevertheless, heating of the sample was carried out so long to ensure that the next compression process will start for the steady state.

The thermal expansion coefficient was determined using samples predefined for static tensile tests. These specimens were fixed at one end while the other end was completely free. As in the case of C-profiles, these samples were painted with paint resistant to high temperatures (up to 1200 °C). The procedure of determination of the coefficient of linear thermal expansion using the DIC system consisted in repeatedly increasing the temperature inside the thermal chamber, waiting for the temperature to stabilize (about 4 min), heating the sample at a specific temperature (10 min), collecting strains at a frequency of 1 Hz for 1 min and then averaging the read strains for each virtual strain gauge. The values of the determined coefficient of linear thermal expansion ranged from 8.6 × 10^−6^ to 10.3 × 10^−6^ [1/K] and the average value of all measurements, which was adopted in further numerical calculations, was 9.2 × 10^−6^ [1/K].

### 3.2. Material Properties

The material properties of titanium at room temperature and at elevated temperatures have been determined according to [42,43], respectively. The value of the Poisson ratio was almost independent of the temperature increase; therefore, this parameter was assumed to be constant. The thermal expansion coefficient was also characterized by a constant relationship between the temperature and the measured strains (cf. Figure 3). The basic material properties of the material under consideration are presented in Table 1.

In the undertaken research, the behavior of the material was modeled as a bilinear and multilinear model. The implementation of the bilinear model requires the declaration of the yield strength, Young’s modulus and tangent modulus (E*). Although the determination of the first two parameters is a relatively simple task, the declaration of the tangent modulus never fully reflects the material characteristics provided by unidirectional tensile tests. In the bilinear approach, the plastic flow curve will either reliably reflect the tensile curve in the range just after reaching the yield point, or it will significantly underestimate the level of stress reached after reaching the yield strength. On the basis of the conducted literature review [39,40,41], the authors concluded that most of the values of the tangent modulus are in the range from one hundredth (0.01 × E) to one thousandth (0.001 × E) of the nominal Young’s modulus. Therefore, in numerical studies carried out with the implementation of the bilinear model, the analyzes were conducted in two ways - separately for different values of the tangent modulus. The yield strength and Young’s modulus values at elevated temperatures are presented in Table 2. Changes in the values of both parameters are non-linear, but tend to be similar to the behavior of aluminum reported by Kaufmann [44].

The declaration of the multilinear model required the transformation of the nominal stress vs. strain curves from uniaxial tensile tests into true stress vs. true strain (logarithmic strain) curves. This transformation was performed according to the Equations (4) and (5):σ_true_ = σ_eng_(1 + ε_eng_)(4)
ε_true_ = ln(1 + ε_eng_)(5)

Exemplary true stress vs. true strain curves are presented in Figure 4. In experimental studies, it was noticed that with increasing temperature not only the value of Young’s modulus and yield strength but also the ultimate tensile strength of the tested material decrease. In addition, at higher temperatures the tensile strength is achieved with lower strains. Moreover, at the temperature of 300 °C the sample suddenly raptured.

Due to the complex behavior of the material at elevated temperatures, it was decided to implement a multilinear material model as a modification of the nominal curve (determined at ambient temperature). Thus, each of the experimental curves determined at elevated temperature (above 23 °C) was described using ten points with the same strains and a modified stress level. The effect of this approach is presented in Figure 4. The inaccuracy of this approach is that it allows for a stress slightly higher than the tensile strength to be achieved. On the other hand, this approach allows a systematic schematization of the nominal tensile curve without the need for additional experimental tests at elevated temperatures. Moreover, the possibility of interrupting the numerical calculations as a result of exceeding the strains or stresses corresponding to the maximum values determined from the experimental tests is excluded. Although the multilinear model allows for a better schematization of the tensile curve, its implementation requires that each point of the curve has a lower true stress/true strain ratio. Thus, it is impossible to reflect the actual course of the curves in the ranges where the stress begins to decrease in relation to the tensile strength. The implemented material model in the multilinear description is included in Table 3.

## 4. FE Model with Assumed Boundary Conditions

Numerical simulations were conducted in Ansys^®^ [42] environment The Newton-Raphson algorithm for the nonlinear stability analysis was employed. The finite element size was set at 2 mm. The prepared model was meshed considering quadratic conventional thin shell elements with 4 nodes, 6 degrees of freedom per node and reduced integration (SHELL181 [45]). The accuracy in modeling of shells is governed by the first order Mindlin–Reissner shear deformation theory. The created numerical model was used in the linear and the non-linear analysis of stability. The nonlinear analysis was divided into three stages: initial compression at ambient temperature (i), heating (ii) and compression at elevated temperature (iii). In all stages of the analysis, a simply supported boundary condition on the supporting and loading edges was modeled. The only difference is that during initial compression and compression to global failure, the loaded edges were able to move along the *Y* direction, while during heating, these displacements were fixed (cf. Figure 5). The compressive load was modeled in such a manner as it was carried on the testing machine, i.e., the loaded edges displace uniformly along the column (uniform shortening of the column). The temperature increase was applied to all nodes simultaneously. This means that possible changes in the temperature distribution with respect to the thickness of the considered profiles were not considered. This is justified by the approximately 2 h heating process, which allowed for the assumption that the state was steady. The discrete model of considered columns with the assumed boundary condition is shown in Figure 5.

Initial linear buckling analysis (LBA) allowed to establish that the considered channel section columns are characterized by a local buckling modes. The first (M1) and second (M2) buckling modes corresponded to two and three buckling half-waves along the length of the analyzed profiles, respectively. Numerically, the relative difference between the first and second buckling loads was less than 3%—a case without temperature influence (Table 4). Such a slight difference may suggest that both cases can coexist in experimental research. This phenomenon has already been the subject of the authors’ research, among others, in works on composite structures [46,47,48]. Moreover, it is noteworthy that no other forms of buckling were identified in the experimental tests either.

Taking into account the results of the preliminary LBA and the influence of modeling the material behavior, it was decided to systematize the approach and carry out numerical calculations in several variants (cf. Table 5). First, the isotropic hardening was defined as multilinear (MULTI) and bilinear (BILI) and in the case of the bilinear model, two values of the tangent modulus (E*) were considered: 0.01 × E and 0.001 × E. With regard to the slight differences in the buckling loads for the first two buckling modes, it was decided to carry out experimental tests in three versions: without initial imperfections (NO IMPER) and with imperfections consistent with the first (M1) or second buckling mode (M2). As part of the implementation of initial imperfections, the imperfection amplitudes at the level of one-tenth (0.1 × t) and one-hundredth (0.01 × t) of the profile wall thickness were assumed. Therefore, the presented approach required fifteen calculation variants for each of the considered temperatures. The diagram of the performed numerical calculations together with the symbols used in the following figures is summarized in Table 5.

## 5. Discussion

The analysis of the results was divided into several subsections, in which the studies concerning only compression without the influence of temperature, the initial compression process, heating to higher temperature and compression at elevated temperatures as well as the variability of the buckling form depending on the temperature applied, were considered separately.

### 5.1. Compression of Channel Section Profiles at Ambient Temperature

According to the diagram in the Table 5, the compression of the considered columns was investigated in fifteen variants, which took into account the influence of the buckling modes and the amplitude of the initial imperfections, as well as a different approach to modeling the behavior of the material itself (bilinear and multilinear models). The force vs. shortening curves of the analyzed columns have been summarized in Figure 6. The first remark concerns the experimental results. It turned out that for three samples tested at ambient temperature, two of them were characterized by buckling mode corresponding to two buckling half-waves along the profile length, while for one sample the buckling mode was three half-waves along the column length. The influence of the buckling mode did not matter much in terms of the load capacity obtained by the column, but it generated slightly different courses of the curves in the range after the profile reached the maximum compressive force. The curves for profiles with two half-waves in the range after reaching the load capacity were characterized by much lower values of the compressive force for the same shortening of the profiles. In the numerical context, the curves that best reflect the experimental paths were determined when implementing the multilinear model. It should be noted that for declarations of initial imperfections consistent with the three buckling half-waves (M2) and without their implementation (NO IMPER), the numerical paths in the postbuckling range almost fit between the experimental paths. Assuming initial imperfections consistent with the first buckling mode (M1), the curves show a slightly greater, compared to experimental tests (EXP_2_DIC and EXP_3_DIC), decrease in stiffness in the range after reaching the load capacity. With regard to the bilinear model, it can be noticed that the conducted analyzes allow for an accurate estimation of the load capacity of the considered profiles and, much worse than the multilinear model, reflect the behavior in the range after reaching the load capacity. It is noteworthy that both when using the bilinear model and the multilinear model, in cases where the buckling mode was two half-waves, the load carrying capacity represented by the profile subjected to compression was slightly higher (but less than 1 kN) than in the case of profiles characterized by three half-waves. A similar relationship was also observed in experimental studies, although the final validation of this statement would require the use of a much larger number of samples. Although in the context of the research undertaken the most important information is the load capacity of the tested profile, after the channel profile reaches the maximum value of the compressive force, all the numerical curves determined with implementation of the bilinear model are below the experimental runs.

An important difference between the experimental and numerical curves concerns the courses of equilibrium paths in the prebuckling range. In order to compare these differences in Table 6 the values of the global stiffness (K) represented by the considered profiles are presented. The K parameter values should be understood as the quotient of the compressive force to the profile shortening, determined in the range from 5 to 15 kN. Based on the value of the K parameter, it can be easily stated that the stiffness induced by numerical tests (regardless of the model) is about three times greater than the stiffness determined from the curves provided by the force and displacement sensor of the UTM. Slightly better results are obtained with the DIC Aramis system. In this case, the stiffness is about two times lower than the stiffness determined by numerical calculations. The DIC system uses the same signal from the force sensor as the UTM. The independence of displacement measurement using the ARAMIS^®^ system suggests the incorrect measurements from the displacement sensor of the UTM. Nevertheless, the registration of the shortening of the profile using the DIC technique is not fully possible, because the points located close to the loaded edges are inside the grooves (cf. Figure 2c) of the test stand and therefore their tracking is impossible. In the undertaken research, the force vs. shortening curves determined with the use of the ARAMIS^®^ system were determined from the points located at the corners of the columns that were closest to the loading support and were simultaneously recorded by both cameras. An exemplary representation of such a point is presented in Figure 7. The vertical arrangement of the ARAMIS^®^ system cameras induced a slight cutting of the test area, but it was a necessary procedure due to the size of the window in the thermal chamber door (cf. Figure 2a) and the dimensions of the profile under consideration. The possibility of registering points located closer to the load (upper) support would allow for a more precise determination of the profile shortening, but at the same time would make it impossible to register the buckling and failure modes of analyzed profiles.

As part of the preliminary analysis, calculations with the implementation of the same initial imperfection amplitudes but with the direction opposite to the nominal buckling modes were also carried out. In the case of the symmetrical mode (two half-waves), no differences were noticed, while in the case of three buckling half-waves, the equilibrium paths were identical in in the range until the load capacity was reached and slightly stiffer (they had their course slightly above the analogous curves with imperfections consistent with the second buckling mode) in the following part. In the context of the performed numerical calculations, it is worth noting that the lack of implementation of the initial imperfections resulted in the presence of the buckling mode characterized by three buckling half-waves for both the bilinear and multilinear model. What is also important, regardless of the amplitude of initial imperfections, the implemented buckling mode did not change during compression (cf. Figure 8), while cases of the buckling mode change during loading can be found, among others, in [46,47,48]. When implementing the multilinear model, the amplitude of the initial imperfections did not generate significant differences in relation to the calculated force vs. shortening curves. In the case of the bilinear model, the influence of initial imperfections was slightly more noticeable. With regard to the Huber–Mises–Hencky equivalent stresses (H-M-H) determined for the points of maximum compressive forces, it can be seen that the multilinear model allowed to achieve stresses much higher than the bilinear model. Moreover, in the case of the multilinear model, the maximum equivalent stresses were slightly higher for cases characterized by two buckling half-waves (over 350 MPa at about 300 MPa for m = 3), while for the bilinear model, the maximum reduced stresses oscillated slightly above the nominal yield strength (cf. Figure 8). Despite some differences in the force vs. shortening curves in the range after reaching the maximum compressive force and taking into account the high compliance of the experimental and numerical results in the context of the induced load-carrying capacities, it was decided to continue the numerical calculations in all indicated variants (cf. Table 5) for each of the analyzed temperatures.

### 5.2. Initial Compression

The first stage of tests involving temperature consisted in initial compression of the column to approx. 5–6% of the nominal load-carrying capacity of the profile. The average values of the initial compressive force for the considered temperatures are presented in Figure 9a. In order to perform numerical calculations and regardless of the considered heating temperature, it was decided to apply the same initial shortening of the profile. In all numerical analyzes, the initial profile shortening was applied at the level of 0.02 mm, which corresponded to a compressive force of 2.1 kN. The Figure 8 also presents a map of equivalent displacements and equivalent stresses corresponding to the assumed level of profile shortening. It is clearly visible that the equiv. displacement distribution corresponds to almost uniform compression and the equiv. stresses are in the range from 6 to 15 MPa. Although the ambient temperature values for each of the experimental tests were different, the average value of the ambient temperature was 22.9 °C (cf. Figure 9b). Thus, in all numerical analyzes, the reference temperature was assumed to be 23 °C.

### 5.3. Heating

After initial compression of the analyzed C-columns, the loading support of the testing machine was blocked and the sample was heated by ΔT. In the tests undertaken, apart from the tests at ambient temperature (approx. 23 °C), the following temperatures inside the thermal chamber were considered: 50 °C, 75 °C, 100 °C, 125 °C, 150 °C, 175 °C, 200 °C, 225 °C, 250 °C, 275 °C and 300 °C. The column heating process for each test took approximately 2 h. The experiment prepared in this way reflected a statically indeterminate system, in which the temperature increase corresponded to the increase of the reaction in the supports. Based on the experimental tests, the reaction forces at the end of the heating process were determined. Their values have been presented in the form of black dots in Figure 10 and compared with the results of numerical calculations.

Based on the performed numerical calculations, it can be concluded that almost all the considered variants of the material models reliably reflect the behavior of the tested channel section columns during heating. It is noteworthy that almost all numerical responses have their course above the values determined from the experiment. Nevertheless, the differences in the registered reaction forces do not exceed 15%. Although most of the applied approaches are relatively correlated, the results obtained for the bilinear model with implemented initial imperfections consisted with the first buckling mode at the level of 0.1 t (data series: BILI_M1_0.1_E*_0.01 and BILI_M1_0.1_E*_0.001) differ slightly from the other results in relation to the heating temperature at the level of 275–300 °C. In both cases, the initial imperfections seem to have a decisive influence on the significant reduction of the reaction value at the end of the heating process. For these cases, however, the relation is maintained that the higher the assumed tangential modulus, the higher the reaction force at the end of the heating process. Up to 100 °C all material models indicate almost identical support reactions. In terms of experiments, the highest reaction force at the end of the heating process corresponds to the temperature of 175 °C, while most often in numerical studies the maximum values were determined for 225 °C. Only when the multilinear hardening characteristic was used without the implementation of initial imperfections (MULTI_NO_IMPER), the highest reaction was achieved, as in the experimental tests, for the temperature of 175 °C. Therefore and bearing in mind the fact that for the tests without the temperature the force vs. shortening curves determined with the application of the multilinear model most accurately reflected the experimental curves, the Figure 11 shows maps of equiv. displacements and equiv. stresses for the subsequent heating phases of the considered channel columns (case: MULTI_NO_IMPER).

Comparing the experimental (cf. Table 7) and numerical (cf. Figure 11) post-heating maps of equivalent displacements, it can be noticed that the flanges of the channel section columns are buckled first and only in the later phase the displacements begin to be more noticeable on the web. The buckling of the flanges of the columns under consideration begins to be clearly visible at the temperature of 175 °C, while in numerical studies it is noticeable already at the temperature of 100–125 °C. The clear difference, however, concerns the fact that the form of buckling in the numerical tests did not change during heating, while in the experimental tests up to the temperature of 225 °C, three buckling half-waves were detected and above this value, only two buckling half-waves were observed. However, it should be borne in mind that each of the maps of equiv. displacements summarized in Table 7 corresponds to a different research sample. A detailed analysis of the experimental displacements for heating cases above 250 °C does not induce the conclusion that in the initial heating stage there were three buckling half-waves and then a change in the buckling mode occurred. This observation, as well as the information about slight differences between the first and second buckling load, suggests the conclusion that the final buckling mode may be determined by the boundary conditions and more precisely the arrangement of the supporting and loaded edges in the test stand grooves (cf. Figure 2c). With regard to numerical calculations, it can be seen that after exceeding the temperature of 100 °C, the web of the channel columns under consideration also buckles. With increasing temperature, the equiv. displacements also increase, while the reduced stresses begin to decrease consistently.

### 5.4. Compression after Heating

The final stage of the conducted research was compression of the columns heated in a statically indeterminate state until the global failure. On the basis of the conducted tests, it can be concluded that in the case of the considered channel columns, heating leads to a reduction in the load-carrying capacity, the greater the higher the temperature increase. It is worth noting that even a temperature increase of 27 °C (50 °C) results in a reduction of the nominal capacity of 3 kN. Further increasing the heating temperature leads not only to lowering the load capacity, but also to the reduction of the shortening value corresponding to the achievement of the maximum compressive force by the profile. From the heating temperature of 200 °C, the maximum compressive force is achieved almost immediately after the compression process starts. For compression at the temperature of 225 °C and 250 °C the load capacity is only slightly higher than the maximum forces determined during heating. This is due to the fact that in these cases the buckling had a thermal nature and occurred before the start of the compression process. Examples of the equilibrium paths of the analyzed profiles are shown in Figure 12. It should be noted that the curves in the Figure 12 have been shifted to the right so that they do not overlap and facilitate the readability of the results obtained.

According to the Figure 12 and the Table 7, it can be seen that for temperatures above 175 °C, the phenomenon of thermal buckling took place. Nevertheless, the compression curves up to the temperature of 225 °C were characterized by a load capacity greater than the maximum reaction force recorded in the heating process (cf. Figure 13a). For compression at 250 °C, the load capacity is almost identical to the maximum reaction force from the heating test, while for compression tests carried out at temperatures above 250 °C, the course of the force vs. shortening curves resembles the character of Figure 13b. In these variants, the maximum force transmitted through the cross-section occurs not during compression but during heating. In the final phase of heating, the reaction force decreases due to thermal buckling and increase of deflections in directions perpendicular to the walls of the analyzed channel columns. Nevertheless, during stage 3 (compression), the maximum recorded compressive force no longer exceeds the maximum value from stage 2 (heating). Although the compression process was not of great importance in these cases, the authors’ aim was also to check the suitability of the prepared numerical models in terms of possible estimation of the load-carrying capacity of compressed structures at elevated temperatures. Figure 14 shows the maximum compressive forces (*P_MAX FEM_*) determined from stage 3—compression after heating—for each of the proposed numerical approaches and Figure 15 presents the results of the calculations related to the results of experimental tests (*P_MAX FEM_/P_EXP_*). The experimentally determined values of the maximum compressive forces are presented in Table 8.

The vertical stripes in the Figure 14 show that different approaches to modeling the material model with or without the implementation of initial imperfections resulted in the estimation of similar values for the maximum compressive forces determined in numerical compression tests at elevated temperatures. On the other hand, the widening of these strips (dark blue and white strips) shows that the increase in temperatures above the thermal buckling temperature did not generate such a regular reduction of the maximum compressive force as it was for temperatures below 175 °C.

The ratio of the maximum compressive forces determined from numerical calculations and experimental tests for stage III—compression at elevated temperature—allows to conclude that in the vast majority of the analyzed cases the results are similar (cf. Figure 15). Assuming the error range at the level of ±5% in relation to the experimental value, it can be seen that almost all the considered approaches allow for the achievement of compliant results. The exception applies only to the cases of using the bilinear model with the implementation of initial imperfections consistent with the first buckling mode and the amplitude at the level of one tenth of the wall thickness of the channel section at temperatures of 275 °C and 300 °C (BILI_M1_0.1_E*_0.01 and BILI_M1_0.1_E*_0.001). In these cases, the maximum compressive force determined by numerical calculations is well below the experimental value. These cases were also characterized by a significant underestimation of the reaction forces in the second stage—heating, compared to the remaining numerical responses. In both of these cases, the initial imperfections seem to have a decisive impact on a significant reduction in the reaction value at the end of the heating process, although the relationship is maintained that the higher the assumed tangential modulus, the higher the value of the maximum compressive force during compression at elevated temperature.

It was decided to finish the analysis of the obtained results with a short discussion of the buckling and failure modes of the analyzed channel columns. At the outset, it can be noticed that clearly visible buckling modes were observed in experimental tests only from the temperature of 175 °C (cf. Table 7). They were m = 3 for temperatures from 175 °C to 225 °C and m = 2 for temperatures from 250 °C to 300 °C, respectively. According to the equiv. displacements shown in Figure 16, the buckling modes did not change during stage III—compression of profiles at elevated temperature. Below the temperature of 175 °C, the maps of equiv. displacements determined from the heating process (DIC Aramis^®^ system) did not allow to unambiguously indicate what mode of buckling will be characterized by further compression of the column. On the basis of the conducted experimental studies, it can be seen that in the range from 50 °C to 150 °C, we have both cases characterized by two and three buckling half-waves. It is mentioned that in none of the experimental scenarios the phenomenon of the change of the buckling mode was noticed both during heating and compression at elevated temperature. This suggests the conclusion that the final mode of buckling in the analyzed cases may be determined by the boundary conditions and more precisely the arrangement of the supporting and loaded edges in the grooves of the test stand.

Although in the experimental tests no changes in the buckling mode during the heating process and the compression after heating were found, in numerical calculations the above phenomenon took place. The phenomenon of changing the nominal buckling mode implemented in the numerical models in the form of initial imperfections concerned only the cases of changing two buckling half-waves into three. The key parameter was the amplitude of the initial deflections. It turns out that assuming the initial imperfections at the level of one hundredth of the wall thickness was insufficient for the heating process above 100 °C (bilinear model) or 125 °C (multiline model) not to change nominally implemented buckling mode. Nevertheless, the assumption of initial imperfections by an order higher (0.1 × t) led to the maintenance of the nominal buckling mode in the full range of the considered temperatures. It is mentioned that the declaration of the tangent modulus at the level of 0.001 and 0.01 did not generate changes in the buckling mode in any of the analyzed cases. Therefore, in Table 9, there is no split into analyzes carried out with different values of the tangent module.

## 6. Conclusions

Within the present study, numerical and experimental investigations of the channel section column subjected to heating and compression at elevated temperature were conducted. The analyzed columns were made of titanium alloy (Grade 2) and simply supported on both ends. The research procedure involved initial compression of the column (i), heating the preloaded column (ii) and compression of the column at elevated temperature to failure (iii). The tests were performed at temperatures from 23 °C to 300 °C. Numerical calculations were carried out in the Ansys^®^ software [42]. The numerical procedures involved the use of two material models: bilinear and multilinear. Within the bilinear model, two values of the tangent modulus were considered. For each of the investigated material models, the influence of initial imperfections consistent with the first or second buckling mode and scenarios without initial imperfections were analyzed. An additional examined parameter was also the amplitude of initial imperfections. Based on the performed numerical and experimental studies, it has been concluded that:the proximity of the first buckling loads may cause the phenomenon of the coexistence of different buckling modes under the same test conditions and lead to a state in which the final buckling mode is determined by slight differences in boundary conditions and more specifically the arrangement of the loading and supporting edges in the test stand grooves;the presence of three buckling half-waves in the case of tests with the use of elevated temperatures resulted in the formation of a plastic mechanism in the middle of the span of the analyzed profiles. This is an advantageous phenomenon because the plastic mechanism is located at a considerable distance from the loading and supporting edges and allows to assume that the assumed boundary conditions are maintained in the full test range. In the case of two buckling half-waves, the plastic mechanism is always formed close to the loading or supporting edges, which means that the boundary conditions, especially in the range after reaching the maximum compressive force, may deviate from the theoretical assumptions;the temperature increase in a statically indeterminate system causes a reduction in the load capacity of the profile, the greater the higher the temperature increase. Already an increase in temperature at the level of 27 °C induces a reduction of the load capacity by 10%, while compression at a temperature of 300 °C reduces the nominal load capacity of the profile by half;for the considered profile, thermal buckling occurs at the temperature of 175 °C. Initially, the flanges buckle, followed by the web of the channel section columns;compression after thermal buckling allows to achieve compressive forces not much higher than those determined in the heating process. In general, the tested columns lose their stability almost immediately after thermal buckling.in the experimental tests, no change in the initial buckling mode was observed, while in the numerical tests the key parameter is the initial imperfection amplitude. Taking this parameter at the level of one-tenth of the wall thickness of the considered profile allows to maintain the nominally assumed buckling mode in the full range of the considered loads. In numerical terms, the change of buckling mode concerned only the change from two to three buckling half-waves along the length of the profiles and it occurred only during heating—stage II;almost all proposed numerical models allow for accurate estimation of both the reaction forces during heating and the maximum compressive forces recorded during compression at elevated temperatures. It also proves the correctness of the determined material characteristics as well as the suitability of shell models for estimating the response of a thin-walled structure subjected to thermomechanical loading;from the perspective of all the results achieved, the best qualitative and quantitative results seem to be achieved with the implementation of the multilinear model. Despite the similar values of the characteristic parameters describing the tests carried out, such as: the reaction force at the end of heating or the maximum force during compression at elevated temperature, the use of the multilinear model also allows for the achievement of force vs. shortening curves in the full load range similar to the experimental ones. In numerical terms, it was also easier to maintain the convergence of the solution using the multilinear model, which in turn also translated into a shorter computational time.

## Figures and Tables

**Figure 1 materials-14-02928-f001:**
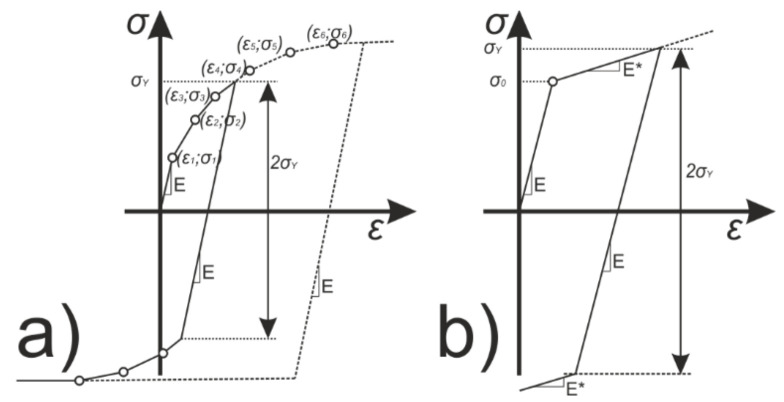
Stress vs. total strain for and multilinear (**a**) and bilinear (**b**) isotropic hardening.

**Figure 2 materials-14-02928-f002:**
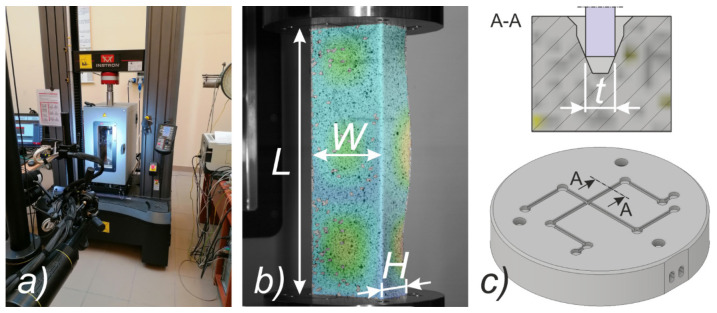
The test stand (**a**) with the considered channel section profile (**b**) and elements for column positioning (**c**).

**Figure 3 materials-14-02928-f003:**
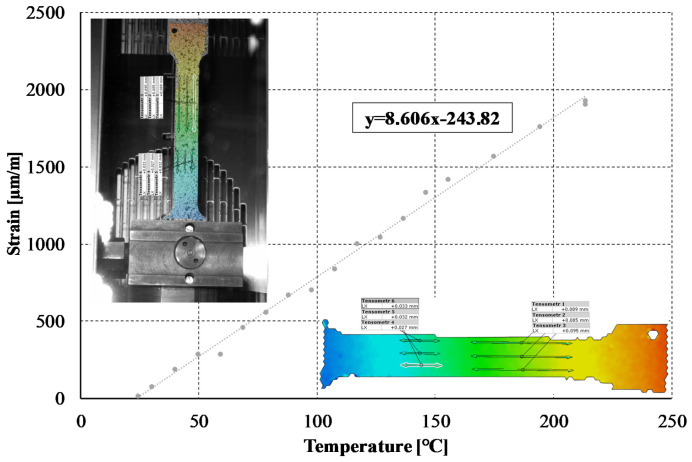
An exemplary strain–temperature increment relationship.

**Figure 4 materials-14-02928-f004:**
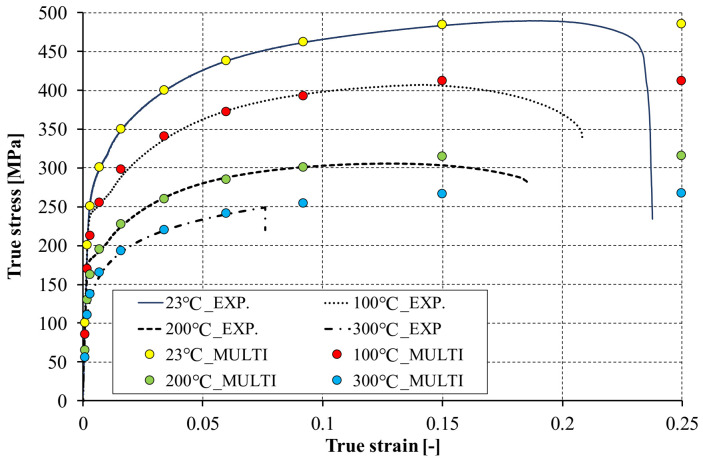
An exemplary true stress vs. true strain curves for considered temperatures.

**Figure 5 materials-14-02928-f005:**
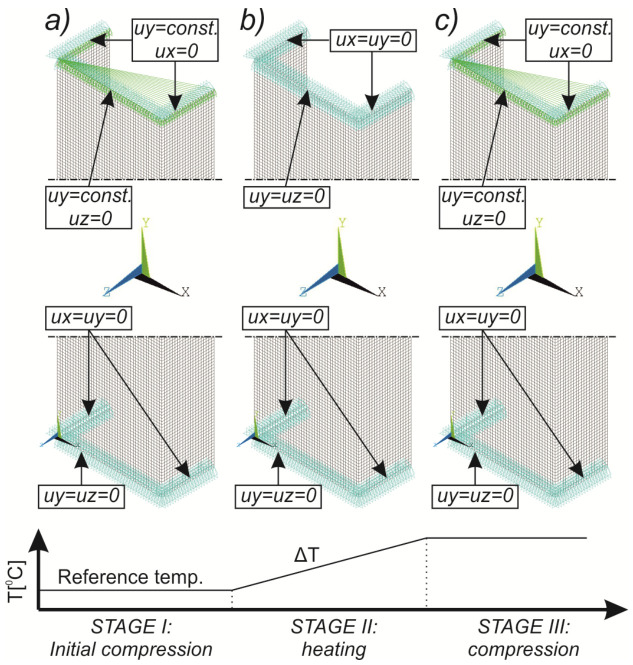
Discrete model of considered columns with the assumed boundary conditions divided into three stages: initial compression (**a**), heating (**b**) and final compression to global failure (**c**).

**Figure 6 materials-14-02928-f006:**
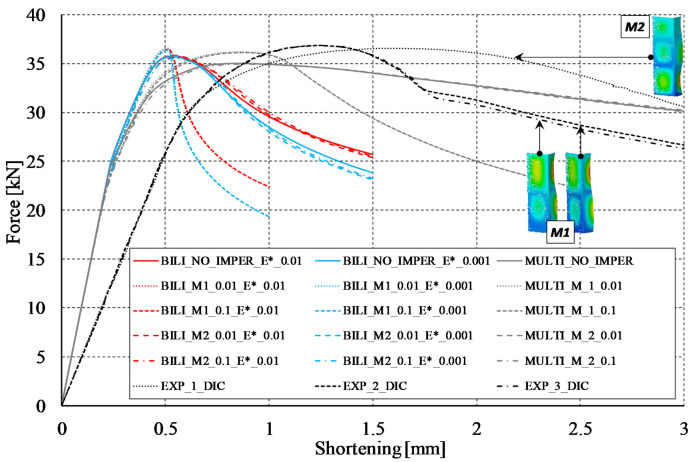
Experimental and numerical equilibrium paths of the considered channel section profile.

**Figure 7 materials-14-02928-f007:**
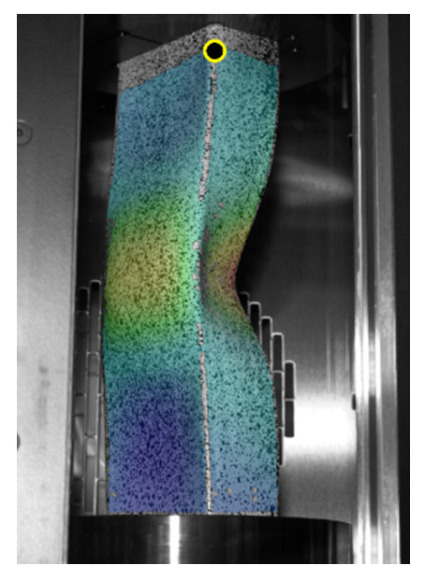
An exemplary registration point for force vs. shortening curves.

**Figure 8 materials-14-02928-f008:**
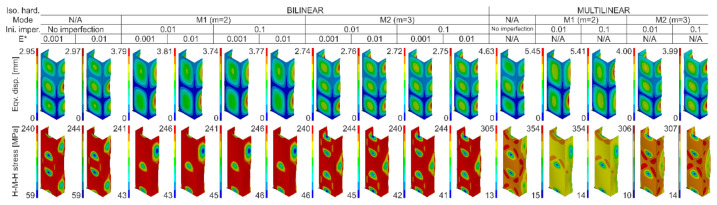
Contour maps of equivalent displacements and H-M-H stresses for maximum compressive forces.

**Figure 9 materials-14-02928-f009:**
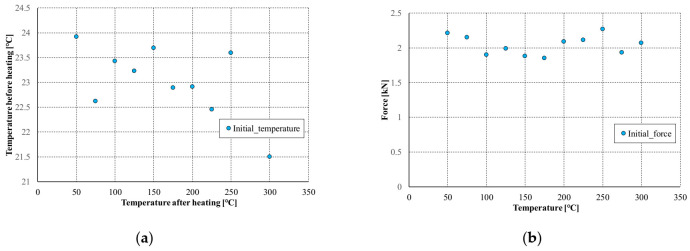
Average values of initial compressive forces (**a**) and ambient temperatures (**b**) before heating.

**Figure 10 materials-14-02928-f010:**
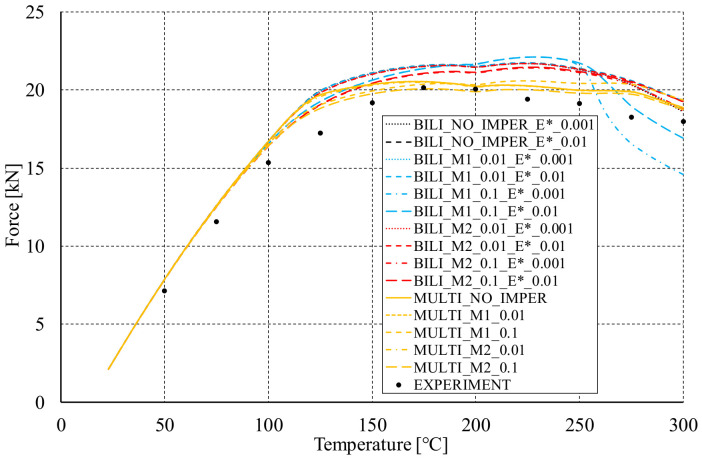
Reaction forces at the end of the heating process.

**Figure 11 materials-14-02928-f011:**
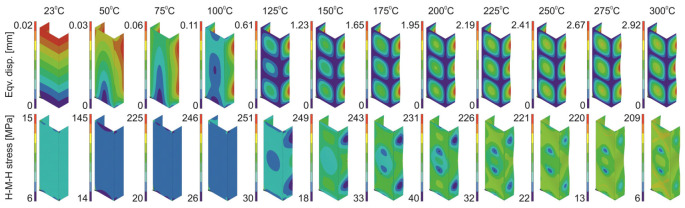
Contour maps of equivalent displacements and H-M-H stresses for heating process.

**Figure 12 materials-14-02928-f012:**
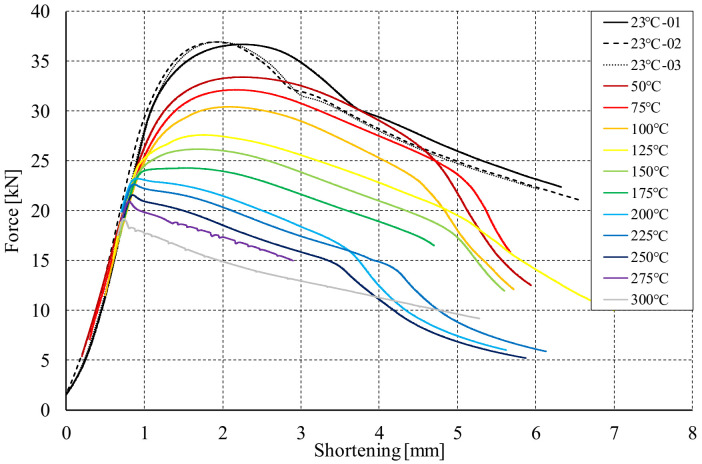
Force vs. shortening curves after the heating.

**Figure 13 materials-14-02928-f013:**
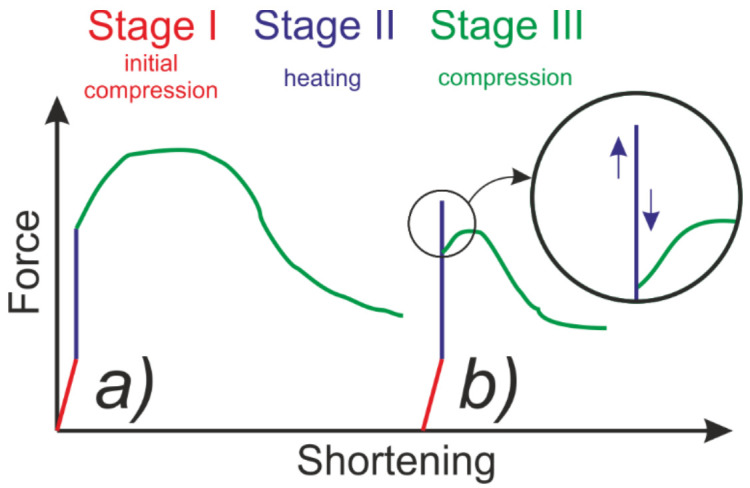
Standard test scheme (**a**) and test scheme with thermal buckling (**b**).

**Figure 14 materials-14-02928-f014:**
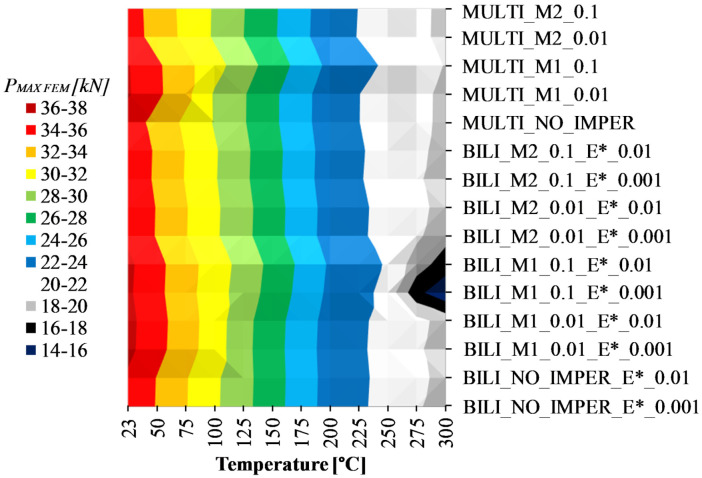
Numerical values of maximum compressive forces from stage III—compression at elevated temperature.

**Figure 15 materials-14-02928-f015:**
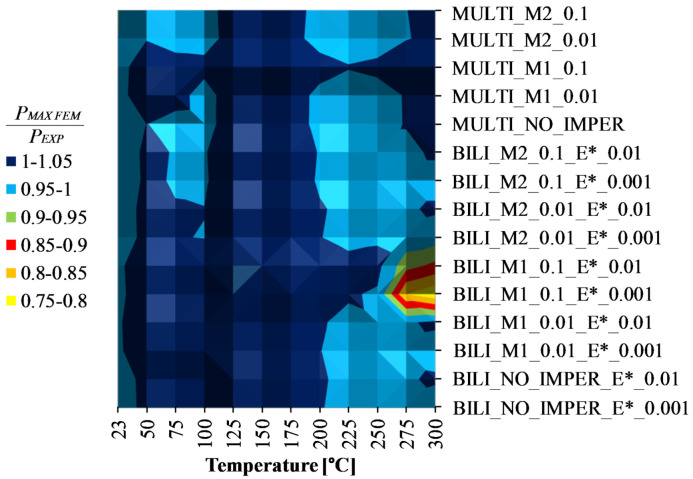
Comparison of numerical and experimental results as *P_MAX FEM_/P_EXP_*.

**Figure 16 materials-14-02928-f016:**
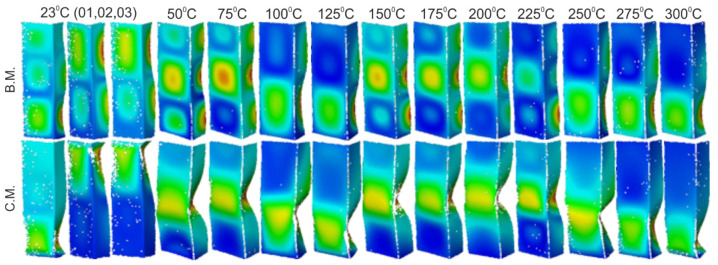
Buckling (B.M.) and collapse modes (C.M.) of considered profiles.

**Table 1 materials-14-02928-t001:** Material properties of considered material.

E [MPa]	ν [-]	α [1/K × 10^−6^]	E* [MPa]
109,300	0.34	9.2	1093 and 109.3

**Table 2 materials-14-02928-t002:** Yield stress [MPa] and Young modulus [GPa] values for different temperatures.

	23 °C	50 °C	75 °C	100 °C	125 °C	150 °C	175 °C	200 °C	225 °C	250 °C	275 °C	300 °C
σ_0_	240	228	216	204	192	180	168	156	150	144	141	132
E	109.3	108.4	107.2	105.7	103.7	101.4	98.7	95.6	92.1	88.3	84.1	79.5

**Table 3 materials-14-02928-t003:** Characteristic points of true stress vs. true strain curves.

	23 °C	50 °C	75 °C	100 °C	125 °C	150 °C	175 °C	200 °C	225 °C	250 °C	275 °C	300 °C
True strain [×10^−2^]	True stress [MPa]											
0.091	100	95	90	85	80	75	70	65	63	60	59	55
0.177	200	190	180	170	160	150	140	130	125	120	118	110
0.227	250	238	225	213	200	188	175	163	156	150	147	138
0.7	300	285	270	255	240	225	210	195	188	180	177	165
1.6	350	333	315	298	280	263	245	228	219	210	206	193
3.4	400	380	360	340	320	300	280	260	250	240	236	220
6	438	416	394	372	350	329	307	285	274	263	258	241
9.2	462	439	416	393	370	347	323	300	289	277	272	254
15	484	460	436	411	387	363	339	315	303	290	285	266
25	485	461	437	412	388	364	340	315	303	291	286	267

**Table 4 materials-14-02928-t004:** Buckling loads (P_cr_) and longitudinal half-wave number (m) of considered columns.

	1st–M1	2nd–M2	3rd
P_cr_ [kN]	24.416	25.091	29.806
m [-]	2	3	4

**Table 5 materials-14-02928-t005:** Considered cases of numerical calculations.

Isotropic Hardening	Bilinear–BILI					Multilinear–MULTI				
Mode:	-	1st [m = 2]–M1		2nd [m = 3]–M2		-	1st [m = 2]–M1		2nd [m = 3]–M2	
Imperfection:	NO IMPER	0.01 × t	0.1 × t	0.01 × t	0.1 × t	NO IMPER	0.01 × t	0.1 × t	0.01 × t	0.1 × t
Tangent modulus–E*	0.001 × E and 0.01 × E	0.001 × E and 0.01 × E	0.001 × E and 0.01 × E	0.001 × E and 0.01 × E	0.001 × E and 0.01 × E	N/A	N/A	N/A	N/A	N/A

**Table 6 materials-14-02928-t006:** Stiffnesses of the profile (K) determined for the case without temperature increase.

	FEM	DIC Aramis^®^	UTM
K [kN/mm]	~105	~52	~33

**Table 7 materials-14-02928-t007:** Post-heating modes of channel section profiles.

50 °C	75 °C	100 °C	125 °C	150 °C	175 °C	200 °C	225 °C	250 °C	275 °C	300 °C
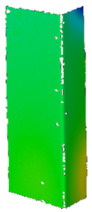	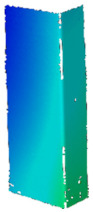	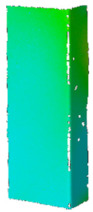	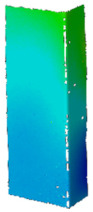	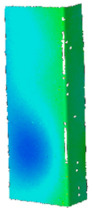	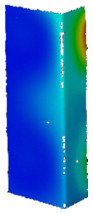	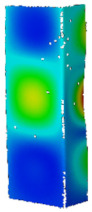	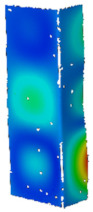	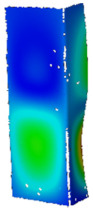	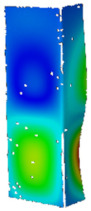	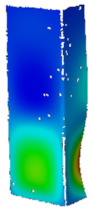

**Table 8 materials-14-02928-t008:** Experimental values of maximum compressive forces [kN] determined from compression at elevated temperature.

23 °C	50 °C	75 °C	100 °C	125 °C	150 °C	175 °C	200 °C	225 °C	250 °C	275 °C	300 °C
36.9	33.3	32.1	30.3	27.5	26.1	24.2	23.2	22.6	21.5	20.7	19.1

**Table 9 materials-14-02928-t009:** Number of buckling half-waves along the column’s length [m].

Isotropic Hardening:	Bilinear					Multilinear					Exp.		
Mode:	-	1st [m = 2]		2nd [m = 3]		-	1st [m = 2]		2nd [m = 3]		N/A		
Imperfection:	-	0.01 × t	0.1 × t	0.01 × t	0.1 × t	-	0.01 × t	0.1 × t	0.01 × t	0.1 × t	N/A		
23 °C	3	2	2	3	3	3	2	2	3	3	3	2	2
50 °C	3	2	2	3	3	3	2	2	3	3	3		
75 °C	3	2	2	3	3	3	2	2	3	3	3		
100 °C	3	2	2	3	3	3	3	2	3	3	2		
125 °C	3	3	2	3	3	3	3	2	3	3	2		
150 °C	3	3	2	3	3	3	3	2	3	3	3		
175 °C	3	3	2	3	3	3	3	2	3	3	3		
200 °C	3	3	2	3	3	3	3	2	3	3	3		
225 °C	3	3	2	3	3	3	3	2	3	3	3		
250 °C	3	3	2	3	3	3	3	2	3	3	2		
275 °C	3	3	2	3	3	3	3	2	3	3	2		
300 °C	3	3	2	3	3	3	3	2	3	3	2		

## Data Availability

The raw/processed data required to reproduce these findings cannot be shared at this time as the data also forms part of an ongoing study.

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
