# Peer review of "Heating and Compression at Elevated Temperature of Thin-Walled Titanium Channel Section Columns"

_materials, 2021, doi:10.3390/ma14112928_

Round 1
Reviewer 1 Report
In this article entitled “Heating and compression at elevated temperature of thin-walled titanium channel section columns”, the authors reported their work on the deformation behavior of thin-walled Ti channel section columns, subjected to three different deformation conditions from room to 300 °C. Both experimental work and numerical simulations were conducted. New results were obtained, correlate with the material stress and temperature. This work is a technically sound and it adds a value to the materials engineering fields. The paper is well written and well organized. I recommend its publication after minor revision as follows.
- As a contribution to a materials journal, the details about the experimental material should be given, such as its chemical composition, manufacture, heat-treatment history, etc. Such information should make this paper comprehensive, without the need to read other literature.
- The conclusions section is too long. It should be much shortened to outline the important conclusions only.
- All symbols should be italicized, such as f, sigma, E, T, etc.
- Leave a space between the number and its unit. The temperature degree is a symbol circle (find it from Symbols), rather than a superscript of the number 0.
Author Response
The authors would like to thank the reviewer for taking his time and reviewing the manuscript, as well as giving a positive evaluation of the submitted article. In response to comments posted by the reviewer, it is mentioned that:
1) A more detailed chemical composition has been included. The text of the publication has been supplemented with the following two sentences: "The specification of the chemical composition of Grade 2 titanium alloy allows the follow-ing standards for the chemical elements: (Ti > 98.9 %, Fe < 0.30 %, O < 0.25 %, C < 0.08 %, N < 0.03 %, H < 0.015 %99.8%). However, the certificate provided by the manufacturer (BIMO TECH [Poland]) indicates the following content of chemical elements in the analyzed material: Fe < 0.018 %, O < 0.14 %, C < 0.0067 %, N < 0.0058 %, H < 0.0022 %, Ti > remainder)."
The remaining parameters, despite the attempt to contact the manufacturer, were not disclosed. Giving additional parameters describing the manufacturing process on the basis of the literature would be somewhat different from the aim and question of the reviewer. According to the e-mail received from the Materials journal, authors must respond to reviewers' questions within 5 days, which unfortunately turns out to be too short to receive more detailed information from the manufacturer.
2) The conclusion section has been shortened by 3 paragraphs. Only the most important information has been left.
3) The names of symbols used in the manuscript are italicized.
4) In the revised version of the text a space has been left between the units and the values ​​of physical parameters. The degree Celsius symbol has been changed throughout the manuscript.
Once again, the authors would like to express their thanks for the effort put into reviewing the manuscript.
Reviewer 2 Report
The paper presents an heating and compression at elevated temperature of thin-walled titanium channel section columns. According to the reviewer’s opinion, the paper is well-structured and clear. The topic is interesting and falls within the aim of the journal. In addition, the results are well-presented and could be helpful to further develop the same topic. Therefore, the paper can be accepted for publication in the current form.
Author Response
The authors would like to thank the reviewer for taking his time and reviewing the manuscript, as well as giving a positive evaluation of the submitted article.
Reviewer 3 Report
See the attached file.

Author Response
The authors would like to thank the reviewer for taking his time and reviewing the manuscript, as well as giving a positive evaluation of the submitted article. In response to comments posted by the reviewer, it is mentioned that:
1) INTRODUCTION (LINES 72 and 73): It is written that the numerical simulations were performed for validating the experimental campaign. In my opinion, the authors should delete this statement, as the numerical models can be validated based on the experimental results, but not the other way around (the experiment is always the reference).
Response 1) The test validation sentence has been modified to "To validate numerical simulations, experimental campaign was performed.".
2) CHAPTER 4 (LINES 281—284): The verb “assumed” is used twice in this sentence; you could consider a review of this part.
Response 2) The sentence has been modified to the following form: "As part of the implementation of initial imperfections, the imperfection amplitudes at the level of one-tenth (0.1 * t) and one-hundredth (0.01 * t) of the profile wall thickness were assumed."
3) SECTION 5.2 (LINE 397): Both are valid, but I suggest abbreviating "equivalent" as equiv. instead of eqv.
Response 3) As proposed by the reviewer, throughout the manuscript, the abbreviation for "equivalent" has been replaced with "equiv". GENERAL
4) COMMENT (E.G., IN LINES 421, 596): In these lines, the authors quantified some discrepancies between results; in my opinion, it would be more appropriate to show them as a percentage instead ofabsolute values.
Response 4) In both sentences the numerical values ​​have been replaced by the relative difference expressed as a percentage.
Once again, the authors would like to express their thanks for the effort put into reviewing the manuscript.